# Interaction of Cyclosporin C with Dy^3+^ Ions in Acetonitrile and in Complex with Dodecylphosphocholine Micelles Determined by NMR Spectroscopy

**DOI:** 10.3390/ijms252413312

**Published:** 2024-12-11

**Authors:** Artyom S. Tarasov, Guzel A. Minnullina, Sergey V. Efimov, Polina P. Kobchikova, Ilya A. Khodov, Vladimir V. Klochkov

**Affiliations:** 1Institute of Physics, Kazan Federal University, 18 Kremlevskaya St., Kazan 420008, Russia; tarasov_as2010@mail.ru (A.S.T.); g.minnullina2010@yandex.ru (G.A.M.); vladimir.klochkov@kpfu.ru (V.V.K.); 2Frank Laboratory of Neutron Physics, Joint Institute for Nuclear Research, 6 Joliot-Curie St., Dubna 141980, Russia; pollymoon@yandex.ru; 3G.A. Krestov Institute of Solution Chemistry, Russian Academy of Sciences, Ivanovo 153045, Russia; iakh@isc-ras.ru

**Keywords:** cyclosporin, NMR spectroscopy, lanthanide, acetonitrile, DPC micelles

## Abstract

The spectral characteristics of cyclosporin C (CsC) with the addition of Dy^3+^ ions in acetonitrile (CD_3_CN) and CsC with Dy^3+^ incorporated into dodecylphosphocholine (DPC) micelle in deuterated water were investigated by high-resolution NMR spectroscopy. The study was focused on the interaction between Dy^3+^ ions and CsC molecules in different environments. Using a combination of one-dimensional and two-dimensional NMR techniques, we obtained information on the spatial features of the peptide molecule and the interaction between CsC and the metal ion. The non-uniform effect of the metal ion on different NMR signals of CsC was observed. The paramagnetic attenuation parameter was calculated for the amide, alpha, and beta protons of CsC upon the addition of Dy^3+^. The metal ion was found to interact with the polar part of the DPC micelle, and the ion also has a significant effect on the NMR signals of amino acid residues from Sar_3_ to d-Ala_8_. This pattern is reproduced in both environments studied here and also agrees with earlier investigations of the CsA–Dy^3+^ complex.

## 1. Introduction

Cyclosporin A (CsA) is widely used to prevent graft rejection in organ transplants due to its immunosuppressive pharmaceutical activity [1]. Cyclosporins can also effectively treat diseases such as dry eye disease, hepatitis B and C, or atopic dermatitis [2,3,4,5]. However, sometimes its use is limited because of severe side effects: administration of cyclosporin may be accompanied by neurotoxicity, nephrotoxicity, hyperkalaemia, hypercalciuria, and other side effects [6,7,8]. So, it is interesting to search for analogues of cyclosporin A with strong immunosuppressive, antiviral, or anti-inflammatory properties among similar molecules, most of which are different from CsA in one or two sites by a change of amino acid or N-methylation state [9,10]. Cyclosporin C is one of those candidates. CsC is a natural analogue of CsA, a cyclic undecapeptide in which aminobutyric acid is replaced by threonine (Figure 1). The results of studies with CsC showed that it has a solid immunosuppressive activity while being less nephrotoxic than CsA [11]. It is known that the activity of cyclosporins depends on their structure and conformation, which the environment can influence: interaction with membranes and micelles, metal ions, or different solvents [12].

The study of the effect of metal ions on biologically active molecules and cell membranes is relevant since the interaction of metal ions with organic compounds can lead to various biological effects through complex formation, covalent bonding, and stabilization of certain conformations. Interactions with ions can also serve as an instrument for studying the structures of macromolecules [13,14]. This work used the dysprosium ion, which belongs to the yttrium subgroup of rare earth metals. It has similar characteristics to other metal ions of the lanthanide group [15]. It is worth noting that in NMR spectroscopy, dysprosium and its complexes are used as a shift reagent [16,17,18,19]. Other metal ions may also form complexes with the peptide and influence the NMR spectra, but we should avoid severe shortening of the *T*_2_ time since it would make recording 2D spectra impossible. Dysprosium was chosen after we had tried several other metals (Co, Mn, and Gd).

The pharmaceutical activity of cyclosporin depends on its membrane permeability, which in turn can vary upon changing the conformation. The formation of cyclosporin–metal ion complexes can significantly affect the structure of the peptide. It is also important to note that cyclosporins bind differently to different metal ions: for example, CsA forms a weaker complex with sodium than with calcium or magnesium [20], and the interaction of CsH with lithium and sodium causes changes in the infrared absorption spectra much weaker than in the case of CsA or CsC [21]. Cyclosporins can act as ionophores and facilitate or inhibit the transport of metal ions across membranes [22]. So, studying how these peptides interact with metal ions can be helpful in understanding why different cyclosporins increase or decrease the rate of metal transport. In cases when the peptide structure is altered negligibly, the paramagnetic effect of the ion on NMR spectra may be a useful tool to study the complex formation.

Previously, other researchers performed experiments on CsA with Dy^3+^ ion in SDS micelles and demonstrated the possibility of forming a stable structure in a micellar solution [23]. Surfactants such as SDS are also capable of interacting with organic molecules and metal ions even below the critical micelle concentration [24], and hence, it is of interest to compare the complex formation process in simple solvents and in micellar solutions.

This work used the DPC micelle to dissolve CsC in D_2_O and mimic the cell membrane. The DPC-based micelle is a spherical aggregate of molecules in which the hydrophilic heads are on the surface and in contact with water, and the hydrophobic tails of the fatty acids of these molecules are inside the sphere and in contact only with each other or with hydrophobic compounds co-dissolved with the lipid. It is usually used as a simplified membrane model (Figure 2) [25,26].

This work aims to study the binding of Dy^3+^ ions with CsC and the possible consequent effect on the peptide structure in different environments such as acetonitrile and membrane-mimicking DPC micelles. The influence of Dy^3+^ ions was characterised by the autoscaled signal volume in the HSQC spectrum, based on the method described in [27,28,29]. Obtained results can be compared in the future with experiments on other metal ions.

## 2. Results

### 2.1. NMR Spectroscopy of Cyclosporin C in Acetonitrile

Analysis of the set of 2D NMR spectra allowed obtaining a total assignment of ^1^H and ^13^C signals for the major conformer of CsC. The chemical shifts of CHα, CHβ, NH/NCH_3_, and carboxyl C’ signals of the main conformer are presented in Table 1. Other conformers with weaker signals were also detected, and TOCSY and HSQC spectra were used to determine the chemical shifts of some signals of the minor conformer (an asterisk * next to the amino acid name indicates whether it belongs to the minor conformer).

The effect of adding dysprosium ion on the HSQC spectrum was examined by comparing HSQC spectra that have been recorded in the absence and the presence of Dy^3+^: the fragment of superposed spectra is shown in Figure 3. Leucine Mle*_4_ CHα signal of the second conformer was not observed, so the fragment of spectra includes CHα signals of 11 residues of the major conformer (written in black letters) and CHα signals of 10 residues of the minor conformer (written in green letters). Unlabelled peaks can also be seen, indicating the presence of other conformers: for instance, the signals of sarcosine Sar_3_ are distinctly visible (Cα at 48.64 ppm and Hα1, Hα2 at 4.92, 3.67 ppm).

Adding dysprosium ions induces a slight broadening of CsC resonances and slightly shifts the peaks. Upon the addition of Dy^3+^ ions, the CHα signals of all residues of the main conformer remain, and the signal of leucine Mle*_9_ of the minor conformer disappears. The signal intensities of these two conformers for free CsC and CsC–Dy^3+^ complex were measured. In the major conformer, the intensity of the signals decreases to about half of those observed in the absence of Dy^3+^ ions, except for leucine residues Mle_9_ and Mle_10_: the integral intensities of these signals obtained in Sparky increase by about 1.5 times. In the minor conformer, the signal intensities also decrease in all residues except Bmt_1_, Sar_3_, Val_5_, and Mle_6_. It should be noted that some peaks of the minor conformer of the CsC–Dy^3+^ complex were not well resolved, and some inaccuracies in the results may be related to this. In the spectral region of CHβ signals, the behaviour of the spectrum after the addition of Dy^3+^ ions is approximately the same: all signals are broadened and the resonances of some residues are insignificantly shifted.

To compare intensities of signals, autoscaled values *V_i_* (*i* is the residue number) were calculated:(1)$$V_{i}=\frac{I_{i}}{\left(1/n\right)\;\sum_{i=1}^{11}I_{i}},$$
where *I_i_* is the measured intensity of the signal, and *n* is the number of residues (11 in CsC). The results for both conformers are shown in Figure 4. In the major conformer before the addition of dysprosium ion, the relative intensity of the signals is approximately the same in all residues of CsC (light blue bars in Figure 4a).

According to Figure 4, adding Dy^3+^ keeps intensities of residues Mle_9_, Mle_10_, and Mva_11_ relatively high, while intensities of other CHα signals calculated using Equation (1) are several times lower and vary slightly. In the minor conformer, relative intensities of CHα signals in all residues change significantly after the addition of the ion and do not show any noticeable trend. These changes in the major conformer of CsC may indicate that Dy^3+^ ions bind to the cyclosporin backbone so that it is placed closer to alpha protons in residues 1, 3, 6, and 7.

Additional tests were carried out with CsC dissolved in dimethylformamide and ions of Gd^3+^ (nitrate) and Mg^2+^ (chloride). One-dimensional ^1^H NMR spectra show subtle changes (Appendix A). Signals slightly shift; their linewidth increases as more Gd^3+^ is added. Generally, NH signals stay in their positions, indicating that the pattern of H-bonds in cyclosporin remains intact. Interestingly, NCH_3_ signals in the central group at 3.3–3.4 ppm are redistributed upon the addition of diamagnetic magnesium salt. This effect may be visible better just due to narrower peaks observed without the paramagnetic compound added.

### 2.2. NMR Spectroscopy of CsC with DPC Micelles in Deuterium Oxide

^1^H NMR spectrum of the CsC–DPC system in D_2_O is presented in Figure 5. The spectrum contains peaks of the DPC micelle (A, B, C, D, E, F, and G, see Figure 2) and several groups of signals from CsC can be distinguished: NCH_3_, CHα, and sidechain groups.

^1^H NMR spectra of CsC–DPC with increasing Dy^3+^ ion concentration in the D_2_O solution were obtained subsequently (Appendix A). When increasing ion concentration, the following changes can be seen in the ^1^H NMR spectrum for the DPC micelle–noticeable broadening of the signals C, F, and E, and shift of the signals A and B towards stronger fields.

Based on the obtained ^1^H NMR spectra in the presence of Dy^3+^ (1 and 2 mM), one can see that CsC is also affected by Dy^3+^ when the peptide is embedded in the micelle [10]. As in the previous section, ^1^H-^13^C HSQC NMR spectra were used to track the changes in the spectra caused by the addition of the lanthanide (Figure 6 and Appendix A).

Complete signals were assigned for CsC with different concentrations of Dy(NO_3_)_3_ in the micellar solution in D_2_O (Table 2 and Appendix A). The initial HSQC spectrum of the CsC–DPC–D_2_O sample was assigned by the previous results [10]. When Dy^3+^ ions were added to the CsC–DPC–D_2_O system, it was found that the DPC micelle and CsC were affected by the surrounding Dy^3+^ ions. Shifts of certain peaks in the HSQC spectrum and additional line broadening can describe this influence. The influence of the ions was characterised quantitatively based on measuring the signal volume in the HSQC spectrum and the paramagnetic attenuation *A_i_* as described before (see Equation (1) [27,28,29]).

Diagrams in Figure 7, Figure 8 and Appendix A show changes in the chemical shifts of NCH_3_, Hα, and Hβ protons upon the addition of dysprosium. The N-methyl signal of Mle_4_ disappears from the HSQC spectra when the ion is added, and chemical shifts of other NCH_3_ resonances increase (Figure 7). Protons of CHα groups show different trends: in residues 1, 2, 3, and 11, signals shift to the low-field region; in residues 4, 9, and 10, to the high-field region (chemical shift decreases), and signals of residues 5, 6, 7, and 8 broaden so much that they become unobservable (Figure 8). For Hβ protons of CsC, the chemical shift in amino acid residues 2, 4, 5, 7, 8, and 11 increases, while in residues 6, 9, and 10, it decreases (Appendix A).

Another way to estimate the effect of Dy^3+^ on the CsC–DPC complex is to calculate the paramagnetic attenuation with increasing Dy^3+^ concentration [27,28,29]. The signal volumes *V_i_* were initially calculated from the ^1^H-^13^C HSQC spectra and normalized by the average volume. The signal volumes without adding Dy^3+^ ions were designated as *V_i_^d^*, and with adding Dy^3+^ ions, *V_i_^p^*. Then the paramagnetic attenuation *A_i_* was calculated using Equation (2) [27], and diagrams of the change in signal volumes and the corresponding paramagnetic attenuations were built (Figure 9, Figure 10 and Appendix A):(2)$$A_{i}=2-\frac{V_{i}^{p}}{V_{i}^{d}}\;\;$$

The *A_i_* value can be used to assume the degree of influence of the ion on the compound. A molecule fragment is more likely to be affected by the ion if the corresponding *A_i_* value differs from 1.0 at a given concentration [28,29]. If *V_i_^p^* < *V_i_^d^*, i.e., paramagnetic signal broadening occurs, *A_i_* increases from 1.0 to 2.0.

Evidently, in the system studied the sequence of amino acid residues from Val_5_ to Dal_8_ is the most susceptible to the presence of the ion. Mle_4_ should probably also be included in this set of residues. Based on the obtained results, a model was built in which the CsC regions affected by Dy^3+^ ions are marked according to their *A_i_* coefficients (see Figure 11). The initial 3D structure underlying this model was taken from our previous studies reported in [10].

## 3. Discussion

Cyclosporins do not belong to the family of metal-binding peptides, typically containing histidine, aspartic acid, glutamic acid, or sulphur-containing amino acids in their composition [30,31]. Nevertheless, they can interact with various metal ions, and this process is reflected in NMR spectra. The peptide–metal complex is expected to have a low binding affinity, so using a lanthanide ion may provide reliable spectral indications of complex formation.

Polar solvents such as DMF, DMSO, or acetone can dissolve cyclosporin, and at the same time, lanthanide salts in these media dissociate into ions. We have chosen acetonitrile due to this fact and also because cyclosporin forms only two or three conformers in this solvent. Signal overlapping is not too severe in this case (unlike the case when cyclosporin is dissolved in DMSO or DMF), which makes it possible to distinguish the conformers and assign NMR signals of at least backbone atoms. Phospholipid micelle, on the other hand, can dissolve cyclosporin in water; the micelle’s interior is where the peptide and ion can stay together and form a more stable complex.

Dy^3+^ exerts a stronger effect on the ^1^H NMR signals of peptides than many other metal ions, especially in terms of signal broadening and shifts [32,33]. This is due to its larger ionic radius and specific coordination chemistry with peptides, which differs from the behaviour of more common metal ions like Zn^2+^ or Mg^2+^ [34,35].

According to our findings, the influence of Dy^3+^ ions on NMR spectra is prominent in amino acids Mle_4_ (NCH_3_ group), Val_5_, Mle_6_, Ala_7_, and d-Ala_8_ (CHα groups), and hence, the ion in the complex reside close to this residue sequence. Coloured spots in Figure 11 demonstrate the positions of atoms whose NMR signals were analysed and revealed the most prominent paramagnetic attenuation.

The structure of the CsC–Dy^3+^ complex dissolved in acetonitrile turns out to be the same, with residues from Mle_4_ to Ala_7_ being close to the ion (and their NMR signals the most broadened), while the backbone site Mle_9_–Mva_11_ is affected the least. The complex in acetonitrile is weaker and appears only for short periods, so the peptide is in exchange between the metal-bound and free forms. Evidently, the exchange-averaged modification of chemical shifts is too small to be visible, but the broadening effect can be observed. The peptide molecule trapped by a micelle already has short *T*_2_ relaxation times and broad NMR peaks. However, the effect of the paramagnetic ion is more substantial and is enough to broaden the peaks even more and shift their resonance frequencies.

Observed spectral changes point to the formation of the peptide–ion complex, but additional arguments are worth finding to reveal whether the structure of the molecule changes. First of all, the chemical shifts of ^13^C backbone atoms are less sensitive to external effects such as solvent composition, while changing the conformation should affect them significantly. Signals in the HSQC spectrum in Figure 3 demonstrate clearly that the amino acid residues differ not only by their δ(^1^H) values but also by δ(^13^C). At the same time, the addition of the lanthanide causes a shift of the peaks mainly along the proton chemical shift axis but does not affect δ(^13^C).

Second, we should clarify how subtle may be differences between two given molecular structures so that we can say that they are two distinct conformers. Cyclosporin’s structure is flexible, especially if we are concerned with side chains. Modifications of the backbone include the formation of *cis*-peptide bonds in different sites. This process is much slower and requires overcoming a high energy barrier, but it can be readily detected in experiments. For example, NOE cross-peaks between adjacent alpha-protons indicate the *cis*-bond, and it is formed in the site Mle_9_–Mle_10_ in the main conformer of CsC (as well as in many other cyclosporins) and in the sites Mva_11_–Bmt_1_ and Thr_2_–Sar_3_ in the second conformer of CsA in CD_3_CN (see Appendix A). Certainly, this type of conformational exchange causes further modifications of the structure; for example, scalar coupling Bmt1(CHα–OHγ) is observed in the major conformer but not in the second form, a strong NOE peak is present for the atom pair Bmt1(CHα–CHβ) in the major form only, etc.

Comparison of 2D NOESY spectra of CsC in the same environment (micelles in water or acetonitrile) recorded with and without the ion shows that the pattern of these NOE peaks does not change. Thus, the general backbone conformation characterised by the distribution of *cis*- and *trans*-peptide bonds is not altered by the metal ion. As an example, two overlayed NOESY spectra recorded in acetonitrile are presented in Appendix A and show a fragment of the NOESY spectrum obtained in the micellar solution in the presence of dysprosium ions. Allowing for the changes in the chemical shifts and a general signal broadening, one can see that the pattern of the NOEs typical of the main conformer (including the Mle_9_Hα–Mle_10_Hα cross-peak) remains the same.

Finally, we also do not expect a substantial influence of the ion on the molecular structure allowing for the dynamic character of the complex formation. Indeed, under stoichiometric conditions (1:4 metal-to-ligand ratio), the interaction between Dy^3^⁺ ions and cyclosporin (CsA) is limited by ligand availability. The stability constant for the CsA–Dy^3+^ complex was found to be *K*_f_ = 10^4^ in [23]. Calculation using KEV software (https://k-ev.org/ accessed on 21 November 2024) [36] shows that at initial concentrations of [Dy^3+^]_0_ = 2.0 mM and [Ligand]_0_ = 1.5 mM with the given stability constant, only 18.75% of Dy^3^⁺ is bound in the complex. This is constrained by the insufficient ligand concentration relative to metal requirements. In the presence of DPC micelles, cyclosporin is likely encapsulated within the micellar environment, reducing its accessibility for direct interaction with Dy^3^⁺. Moreover, Dy^3^⁺ ions may preferentially interact with the hydrophilic micellar surface rather than with cyclosporin. Consequently, the impact of Dy^3^⁺ on cyclosporin is minimal under these conditions, particularly in the presence of DPC micelles. Without considering micelle effects, the maximum concentration of the peptide–metal complex is limited to 0.375 mM, determined by the available ligand.

Our results agree with those reported in [23] for the CsA–Dy^3+^ complex. Furthermore, vibrational circular dichroism spectroscopy of the CsC–Mg^2+^ complex [37] also shows that the complex is formed without disrupting the intramolecular network of hydrogen bonds, which also may happen without significant conformational changes. In addition, we demonstrated that acetonitrile can be a convenient solvent for investigating the interaction of cyclosporin and metal, including the cases of the selective interaction of certain conformers with the ion.

## 4. Materials and Methods

NMR measurements were carried out on a Bruker Avance III HD 700 spectrometer (700 MHz for ^1^H, 175 MHz for ^13^C). All samples were prepared in standard 5-mm NMR tubes. The solution volume was 0.6 mL; the temperature of 298 K was stabilized during the experiments.

Cyclosporin C (CSC) was purchased from AvaChem Scientific (San Antonio, TX, USA). CsC in the CD_3_CN samples were prepared by dissolving CsC in deuterated acetonitrile (CD_3_CN, 99.8%) at a concentration of 1.4 mM. Dysprosium nitrate Dy(NO_3_)_3_·6H_2_O was dissolved in acetonitrile, and some amount of this stock solution (<0.02 mL) was added to the NMR sample. Dy^3+^ ions were added to achieve the relative ion:peptide molar ratio of 1:4. Two-dimensional spectra (DQF-COSY, TOCSY, ROESY, ^1^H-^13^C HSQC, and ^1^H-^13^C HMBC) were recorded before adding the ion, afterward the ^1^H-^13^C HSQC spectrum of the CsC–Dy^3+^ complex was obtained. Correlation spectra were obtained in the echo/anti-echo acquisition mode (pulse programs mlevetgp, hsqcetgpsi, and hmbcetgpnd); nuclear Overhauser effect spectra were recorded using programs noesygpph or roesyph.2.

The CsC-DPC mixture was dissolved in deuterium oxide (D_2_O from Solvex-D, 99.8%; DPC from Avanti Polar Lipids, Alabaster, AL, USA). The concentration of peptide was 1.5 mM; the concentration of DPC was 46.6 mM. A series of one-dimensional ^1^H and two-dimensional ^1^H-^13^C HSQC NMR spectra were acquired with two Dy(NO_3_)_3_ concentrations in the probe (1.0 and 2.0 mM).

Data acquisition and processing were carried out using TopSpin 3.5. Homo- and heteronuclear 2D NMR techniques were used to assign the signals of CsC in CD_3_CN and CsC–DPC in D_2_O without the addition of the ion, and another HSQC spectrum was used for signal assignment of CsC–Dy^3+^ complex. The spectral analysis and peak integration in HSQC spectra were made with the aid of the Sparky program [38]. Overlapping signals were not taken into account in calculating volumes to avoid an incorrect calculation of the paramagnetic attenuation.

## Figures and Tables

**Figure 1 ijms-25-13312-f001:**
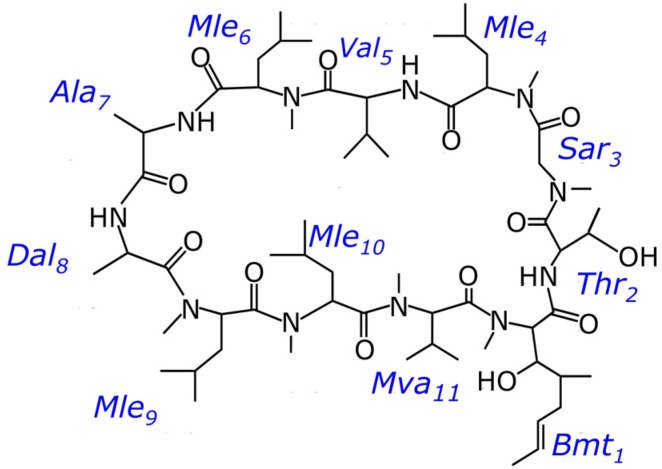
Chemical structure of cyclosporin C.

**Figure 2 ijms-25-13312-f002:**
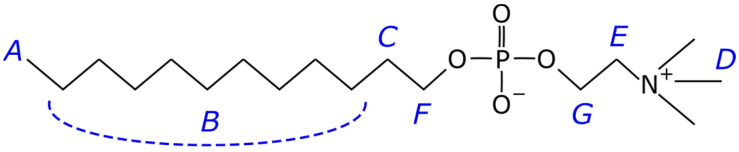
Chemical structure of dodecylphosphocholine (DPC). Letters from A to G label the atoms in accordance with the spectrum in Appendix A; signal B unites several methylene atoms joined by the dashed line.

**Figure 3 ijms-25-13312-f003:**
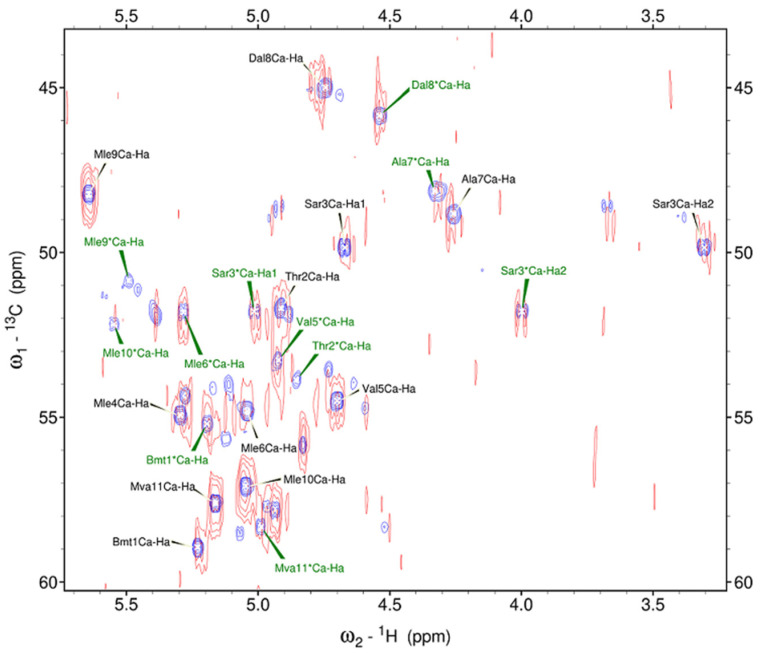
Fragment of ^1^H-^13^C HSQC spectra (700 MHz, 298 K) of CsC in CD_3_CN with and without the addition of Dy^3+^: purple peaks correspond to the spectrum recorded with the pure CsC sample; red peaks, to the spectrum of the CsC–Dy^3+^ mix. * indicates the minor conformer.

**Figure 4 ijms-25-13312-f004:**
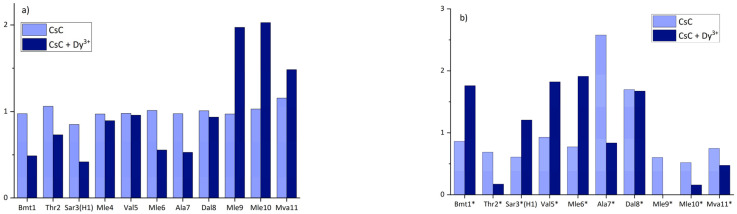
Relative intensities of CHα signals for the (**a**) major and (**b**) minor conformer of CsC in CD_3_CN before and after adding Dy^3+^. Asterisk near the amino acid labels in panel (**b**) remind that they belong to the second conformer.

**Figure 5 ijms-25-13312-f005:**
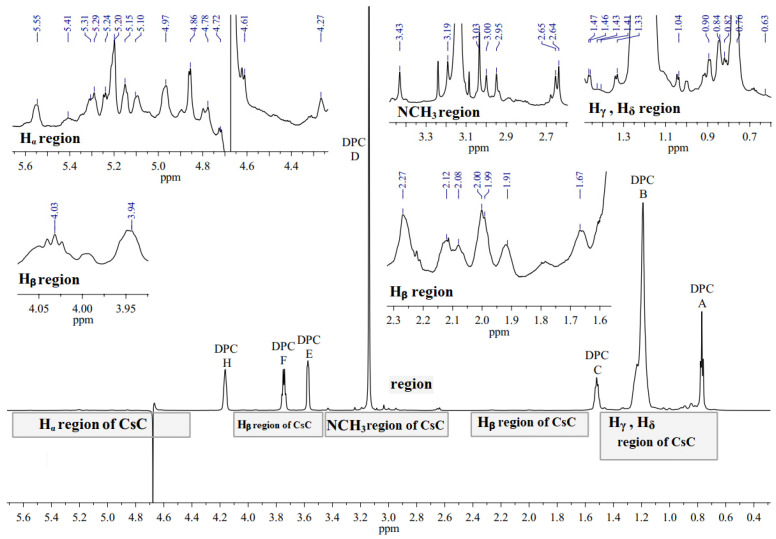
^1^H NMR spectrum of CsC–DPC complex without Dy(NO_3_)_3_ in D_2_O (700 MHz, 298 K). Signals of DPC are labelled by capital Latin letters.

**Figure 6 ijms-25-13312-f006:**
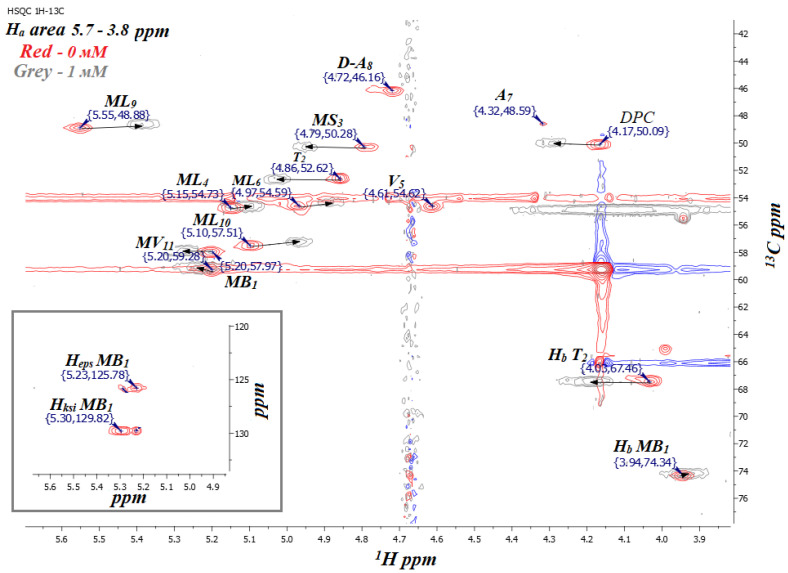
Fragment of ^1^H-^13^C HSQC NMR spectrum of CsC–DPC complex with different concentrations of Dy(NO_3_)_3_ (red, 0 mM; grey, 1 mM Dy^3+^) in D_2_O (700 MHz, 298 K).

**Figure 7 ijms-25-13312-f007:**
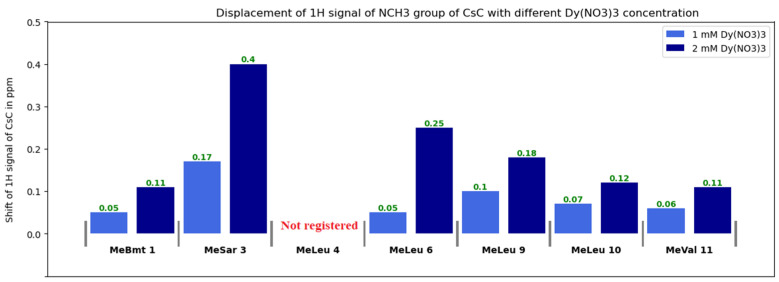
Changes in ^1^H chemical shifts δ(NCH_3_) of CsC with different Dy(NO_3_)_3_ concentrations.

**Figure 8 ijms-25-13312-f008:**
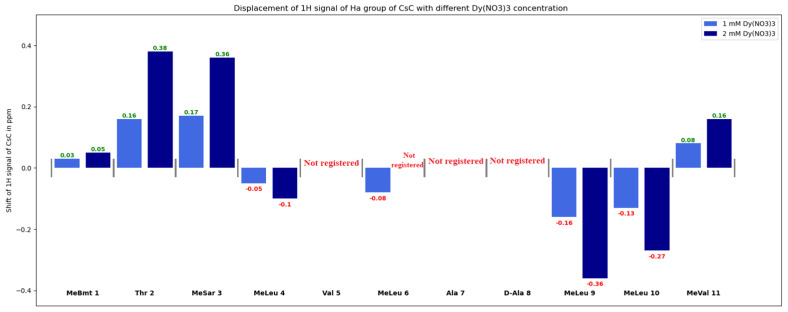
Changes in δ(H_α_) group of CsC with different Dy(NO_3_)_3_ concentrations.

**Figure 9 ijms-25-13312-f009:**
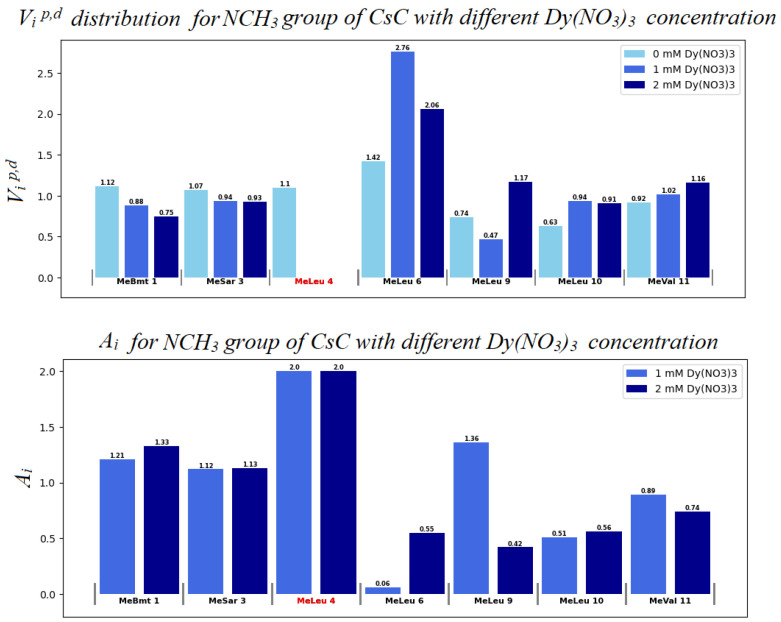
Distribution of autoscaled HSQC volume signals *V_i_^p,d^* for NCH_3_ groups of CsC. Light blue bars (in the upper panel) correspond to the absence of Dy^3+^ (*V_i_^d^* values); blue bars to 1 mM Dy^3+^ (*V_i_^p^*); and dark blue bars to 2 mM Dy^3+^. Some signals disappear so that we assume that for them *A* = 2, here and in Figure 10 corresponding residues are marked by red colour.

**Figure 10 ijms-25-13312-f010:**
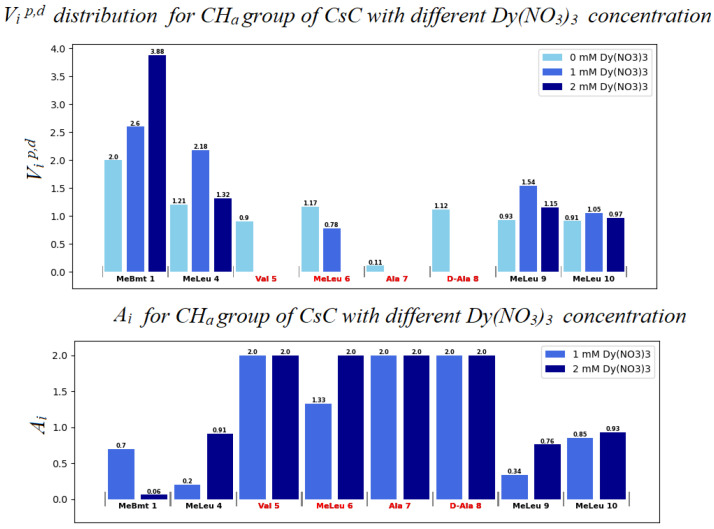
Distribution of autoscaled HSQC volume signals *V_i_^p,d^* for alpha protons of CsC. Light blue bars (in the upper panel) correspond to the absence of Dy^3+^ (*V_i_^d^* values); blue bars to 1 mM Dy^3+^ (*V_i_^p^*); and dark blue bars to 2 mM Dy^3+^.

**Figure 11 ijms-25-13312-f011:**
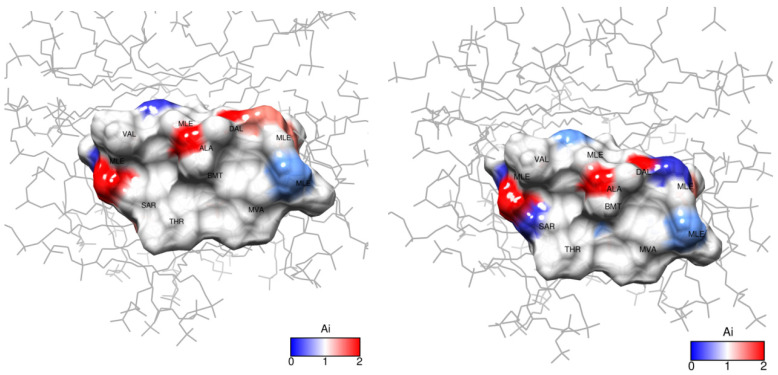
Paramagnetic attenuation effect of Dy^3+^ ions on cyclosporin C within DPC micelle in D_2_O: (**left**), 1 mM, (**right**), 2 mM Dy(NO_3_)_3_. *A_i_* is the attenuation factor given by Equation (2).

**Table 1 ijms-25-13312-t001:** ^1^H and ^13^C chemical shifts (ppm) of CsC in major conformer in acetonitrile (CD_3_CN). Residue numbering corresponds to Figure 1. Underline in two columns for NCH_3_ indicate the nucleus for which the chemical shift is given.

	Cα	Hα	Cβ	Hβ	NH	NCH_3_	NCH_3_	C’
Bmt1	58.94	5.229	73.15	3.925		32.91	3.378	170.0
Thr2	51.70	4.912	67.07	3.977	7.773			173.1
Sar3	49.85	4.670; 3.305				38.73	3.326	171.2
Mle4	54.96	5.302	35.59	1.828; 1.627		30.77	3.009	169.6
Val5	54.50	4.699	31.62	2.210	7.297			173.5
Mle6	54.80	5.042	36.88	2.007; 1.286		30.98	3.125	171.1
Ala7	48.83	4.260	15.55	1.339	7.458			170.8
Dal8	45.03	4.748	17.39	1.187	7.265			173.6
Mle9	48.26	5.648	38.69	2.049; 1.208		29.14	3.082	170.1
Mle10	57.08	5.047	40.64	2.049; 1.191		29.31	2.652	169.8
Mva11	57.60	5.162	28.77	2.114		29.58	2.655	173.2

**Table 2 ijms-25-13312-t002:** ^1^H chemical shifts of CsC with DPC micelles in D_2_O at 298 K. Chemical shifts with 1 mM Dy^3+^ added are listed in parentheses ( ); with, 2 mM Dy^3+^, in square brackets [ ].

	Hα	Hβ	Hγ	Hδ	NCH_3_
Bmt1	5.20 (5.23) [5.25]	3.94 (3.94) [3.92]	1.50 (1.38) [1.21]	0.75 (0.62) [0.46]	3.43 (3.48) [3.54]
Thr2	4.86 (5.02) [5.24]	4.03 (4.20) [4.38]	1.04 (1.22) [1.43]		
Sar3	4.79; 3.52 (4.96; 3.70) [5.15; 3.90]				3.19 (3.36) [3.59]
Mle4	5.15 (5.10) [5.05]	1.67; 1.57 (1.70; 1.61) [1.76; 1.64]	1.40 (1.37) [1.35]	0.85 (0.83; 0.61) [0.43; 0.15]	2.94 (n/d) *
Val5	4.61 (n/d) *	1.99 (2.12) [2.28]	0.89; 0.81 (0.60; 0.49) [0.22; 0.12]		
Mle6	4.97 (4.89) [n/d] *	2.00; 1.17 (1.87; 0.97) [1.68; 0.72]	1.47 (1.51) [1.50]	0.84; 0.82 (0.60) [n/d] *	3.03 (3.08) [3.11]
Ala7	4.32 (n/d) *	1.33 (1.46) [1.62]			
Dal8	4.78 (n/d) *	1.15 (1.18) [1.27]			
Mle9	5.55 (5.39) [5.19]	2.08; 1.06 (1.99; 0.86) [1.88; 0.61]	1.32 (1.33) [1.31]	0.89; 0.81 (0.60; 0.49) [0.22; 0.12]	3.00 (3.10) [3.18]
Mle10	5.10 (4.97) [4.83]	2.27; 0.90 (2.03; 0.62) [1.75; 0.28]	1.43 (1.43) [1.44]	0.62 (0.69) [0.74]	2.63 (2.70) [2.75]
Mva11	5.20 (5.28) [5.36]	2.12 (2.28) [2.44]	0.84; 0.79 (1.05; 0.96) [1.26; 1.16]		2.65 (2.71) [2.76]

* Signal disappears upon further addition of the salt.

## Data Availability

The raw data supporting the conclusions of this article will be made available by the authors on request.

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
