# Peer review of "Interaction of Cyclosporin C with Dy3+ Ions in Acetonitrile and in Complex with Dodecylphosphocholine Micelles Determined by NMR Spectroscopy"

_ijms, 2024, doi:10.3390/ijms252413312_

Round 1

Reviewer 1 Report

Comments and Suggestions for Authors

NMR is a great tool to investigate weak interactions such as peptide-metal ion interaction. The authors discussed cyclosporin C (CsC) interaction with dysprosium(Dy3+) ions in two different environments. They assumed that peptide-metal ion interactions are important to understand the biological effects. However, the reviewer can’t see any novelty of the work. The manuscript just showed the chemical shifts change (broadening) of CsC with Dy3+ and DPC micelle. They didn’t calculate the structure of CsC. The major and minor conformations of CsC are also ambiguous. So, the manuscript is not suitable for the publication in IJMS. The following are the suggestions to improve the manuscript.

1.Since peptide-metal complex is expected to have a low binding affinity, they used a lanthanide ion to get some information of complex formation, which assumed that metal ion-binding of CsC might be the same regardless of metal species. To confirm this assumption, the authors may perform additional experiments. For example, they may do the titration experiment for the complex of (CsS and Dy3+) sample with any salt solution to see whether the signals of CsS may be back. This experiment may show that Dy3+ ion competes with other metal ions.

2. What are the major and minor conformations? The authors should get the NOEs of the sample to determine the structures of CsC. The authors assumed that previously determined structures of CsC might be the same (Fig. 11) although the environment is different. That was a reason why the reviewer couldn’t find the novelty.

3. The authors need to compare the previous results [Ref. 10] and current work.

4. Since the experiments the authors presented are not complicated, it would be interesting to see whether other cyclosporin variants are the same or different in this environment. 

Author Response

Comment 1. Since peptide-metal complex is expected to have a low binding affinity, they used a lanthanide ion to get some information of complex formation, which assumed that metal ion-binding of CsC might be the same regardless of metal species. To confirm this assumption, the authors may perform additional experiments. For example, they may do the titration experiment for the complex of (CsS and Dy3+) sample with any salt solution to see whether the signals of CsS may be back. This experiment may show that Dy3+ ion competes with other metal ions.

Response 1. Here we used two consecutive additions of Dy3+ to check if the trend of NMR signal shifting is present. Of course, this is not a proper titration series, but increasing the salt concentration too much will male NMR signals unobservable due to paramagnetic broadening. Regarding interaction with other ions: this would be interesting, but not all ions lead to prominent spectral changes. We have checked also gadolinium, cobalt, manganese – the effect was either negligible or resulted in signal broadening only, without changes in the chemical shift.
Maybe mentioning this fact will be useful, so we extended paragraph 2 of the Introduction (lines 52-55).

Comment 2. What are the major and minor conformations? The authors should get the NOEs of the sample to determine the structures of CsC. The authors assumed that previously determined structures of CsC might be the same (Fig. 11) although the environment is different. That was a reason why the reviewer couldn’t find the novelty.

Response 2. The major and minor conformations differ, first of all, by distribution of trans- and cis-bonds in the backbone. Improvements in the article regarding this comment are described further in response 3.

Comment 3. The authors need to compare the previous results [Ref. 10] and current work.

Response 3. A question similar to both these questions 2 and 3 was also made by Reviewer 2. Indeed, possible change in the structure upon metal binding would be an interesting finding; as well as the fact that the structure can bind the ion and remain its conformation. We added new experimental material: namely, NOESY (ROESY) spectra of CsC in acetonitrile and in micelles with and without the ion are compared in the Supplementary materials (Figures S5, S6, S7). They are described in the Discussion (line 254 and below), and together with theoretical estimates of complex stability prove that in the case of cyclosporin C (major conformer) and Dy3+ the binding to the ion does not lead to a conformational change in the peptide backbone.

Comment 4. Since the experiments the authors presented are not complicated, it would be interesting to see whether other cyclosporin variants are the same or different in this environment.

Response 4. Together with ref. [23] we have comparison of CsA and CsC. The experiments are not unique, but now now we see that more planning is needed (including comparison with other studies which have been reported – optical, DFT calculations, etc.) to choose the most promising pair of peptide and ion. A work in this direction continues in our laboratory.

Reviewer 2 Report

Comments and Suggestions for Authors

v

Author Response

No comments are given.

Reviewer 3 Report

Comments and Suggestions for Authors

This manuscript describes metal-ion interactions of a medically relevant molecule (cyclosporin C) using NMR spectroscopy. The authors use a metal with high magnetic strength (dysprosium) and look at chemical shift changes due to the presence of metal ion to find out where the metal ion is preferably associated within cyclosporin C. The authors do this in either acetonitrile or micelles in deuterium oxide. This methodology is well established.

It can be debated how far these results are biologically and medically relevant and the authors may want to include some additional comments about this in their manuscript. As the authors accurately pointed out, cyclosporins do not belong to the family of metal-binding peptides. Furthermore, there is no known biological role of dysprosium, although its frequent use in industry mostly for its magnetic properties may make it relevant as a contaminant. The biological presence of dysposium in membranes is limited and likely far below the concentrations that were used in the experiment described in this manuscript.

The authors state that the structure of cyclosporin is not significantly effected by the presence of dysposium. They should show evidence and describe the relevant calculations. As is, it appears that the authors describe the chemical shift changes but these are to be used in some kind software modelling program to actually calculate the structure.

Some specific issues:

Line 196-198: 'Based on the obtained results, a model was built in which the CsC regions affected by the Dy3+ ion are marked according to their Ai coefficients (see Fig. 11). The initial 3D structure underlying this model was taken from our previous studies reported in [10].': It appears that the authors simply used the structure without metal ion and then assumed that the structure won't change upon interaction with the metal. This cannot be assumed. The authors should make the relevant calculations and if they actually did them, they should be reported in detail.

Line 209: 'too many': Please quantify.

Lines 218-219: 'The structure of the CsC–Dy3+ complex dissolved in acetonitrile turns out to be the same': How do the authors know this? Did they do the relevant calculations? If so, these should be reported.

Line 229-230: 'Other cyclosporin variants and analogues can form two major conformers with similar populations in media based on this solvent': Please include reference(s).

Line 234-254: The source of the chemicals and the isotopic purity of deuterated acetonitrile should be added. Furthermore, the purity and the form of dysprosium ions should be reported. Also, the pulse programs that were used to acquire two-dimensional spectra should be included.

Line 254: 'which led to an incorrect calculation of the paramagnetic attenuation.': Probably, the authors wanted to convey that they wanted to avoid an incorrect calculation of the paramagnetic attenuation. Please rephrase if that is the case.

Lines 261-262: Delete: 'For research articles with several authors, a short paragraph specifying their individual contributions must be provided. The following statements should be used “'

Author Response

Ionic radius and charge are basic properties determining their interaction with macromolecules. So, different metals can mimic other ones, if their radii are similar, but lanthanides in this case also offer the ability to introduce prominent changes in NMR spectra. In cases when the peptide structure is altered negligibly (this seems true for CsC - we will return to this question in the responses below), paramagnetic effect of the ion on NMR spectra may be a useful tool to study the complex formation.

Comment 1. Line 196-198: 'Based on the obtained results, a model was built in which the CsC regions affected by the Dy3+ ion are marked according to their Ai coefficients (see Fig. 11). The initial 3D structure underlying this model was taken from our previous studies reported in [10].': It appears that the authors simply used the structure without metal ion and then assumed that the structure won't change upon interaction with the metal. This cannot be assumed. The authors should make the relevant calculations and if they actually did them, they should be reported in detail.
Response 1. The structure was not recalculated to fit the ion. Indeed, the question of the reviewer is important, since the aim of the article was stated as “to study … possible consequent effect on the peptide structure.” We cannot say a priori if it changes or not; it seems that different scenarios are possible with different cyclosporins, as was revealed by Bodack and co-authors by optical methods in the paper “Solution conformations of cyclosporins and magnesium-cyclosporin complexes determined by vibrational circular dichroism” [https://dx.doi.org/10.1002/bip.10513]. However, there are indications of the fact that the structure (at least the backbone) does not alter; this is related to question 3 (see below) and considered in the Discussion section.

Comment 2. Line 209: 'too many': Please quantify.
Response 2. 2-3 instead of 6 or more. We’ve rewritten the sentence.

Comment 3. Lines 218-219: 'The structure of the CsC–Dy3+ complex dissolved in acetonitrile turns out to be the same': How do the authors know this? Did they do the relevant calculations? If so, these should be reported.
Response 3. This comment is close to question 1 of Reviewer 1. There are several arguments based on NMR observations and also on some theoretical assumptions about the complex stability. We have not performed simulations of the complex, but extended the Discussion section to clarify this question.

Comment 4. Line 229-230: 'Other cyclosporin variants and analogues can form two major conformers with similar populations in media based on this solvent': Please include reference(s).
Response 4. This particular note looks too general. In fact, it is based on some of our unpublished yet results, so it’s better to remove this sentence. The possibility of selective interaction is still mentioned.

Comment 5. Line 234-254: The source of the chemicals and the isotopic purity of deuterated acetonitrile should be added. Furthermore, the purity and the form of dysprosium ions should be reported. Also, the pulse programs that were used to acquire two-dimensional spectra should be included.
Response 5. Information added. Dysprosium salt was provided by colleagues from the chemical department.

Comment 6. Line 254: 'which led to an incorrect calculation of the paramagnetic attenuation.': Probably, the authors wanted to convey that they wanted to avoid an incorrect calculation of the paramagnetic attenuation. Please rephrase if that is the case.
Response 6. Corrected.

Comment 7. Lines 261-262: Delete: 'For research articles with several authors, a short paragraph specifying their individual contributions must be provided. The following statements should be used “'
Response 7. Corrected.

Round 2

Reviewer 1 Report

Comments and Suggestions for Authors

The revised manuscript shows the improvement. The following is a minor question. 

The authors should compare the effect of Dy3+ and other metal ions based on the 1D spectra of CsC to show how other metal ions affect the 1H signals of peptides. This will help to understand why different cyclosporins increase or decrease the rate of metal transport. 

Author Response

Comments 1:The revised manuscript shows the improvement. The following is a minor question. 
The authors should compare the effect of Dy3+ and other metal ions based on the 1D spectra of CsC to show how other metal ions affect the 1H signals of peptides. This will help to understand why different cyclosporins increase or decrease the rate of metal transport.

Response 1: This is a wide and interesting question. Addition of a paramagnetic metal ion may lead to NMR signal shift or broadening, depending on the properties of the ion. At the same time, the complex formed can possess different stability and structure.
According to our experience, addition of transition metal ions such as cobalt or manganese leads to signal broadening (checked with CsA in micellar solutions and with CsE dissolved in DMF). Gadolinium also leads mainly to signal broadening, while dysprosium causes changes in the chemical shift (observed not only on CsC, but also on CsB and CsE). It is also known that ions such as magnesium or alkali metals can also modify NMR spectra, reflecting possible structural changes.
We decided to make some additional tests with CsC dissolved in dimethylformamide and ions of Gd3+ (nitrate) and Mg2+ (chloride). There are multiple conformations coexisting in this solvent, so the ion has a possibility to interact with suitable ones. One-dimensional 1H NMR spectra show, however, that the changes are negligible. Signals slightly shift; their linewidth increases as more Gd is added. Broadened solvent signals hamper observation of neighbor peaks of the peptide. Generally, NH signals stay on their positions, indicating that the pattern of H-bonds in cyclosporin remains intact. Interestingly, NCH3 signals in the central group at 3.3 – 3.4 ppm are redistributed upon addition of diamagnetic Mg chloride. This may be attributed to structural changes; we should be aware, though, that some amount of crystalline water may be present in the sample after salt addition. Note also that other peaks, including two rightmost NCH3 signals, stay in their position with both ions checked here.
New spectra are added to the supporting information file; description in the main text appear in the end of paragraph 2.1 (lines 160-166) and Disussion (paragraph 3, lines 250-253).

Reviewer 3 Report

Comments and Suggestions for Authors

The manuscript significantly improved after revision.

It is understood in the field of NMR spectroscopy, that the presence of minor peaks in the spectra is often attributed to minor conformations which are often not taken into consideration. Thus, it is good to see the authors comment on 2 conformations. However, the authors also mention the possible presence of a third dominant conformer (lines 236-238):' We have chosen acetonitrile due to this fact and also because cyclosporin forms only two or three dominant
conformers in this solvent', although this seems to be in contradiction to Lines 265-266. 'Second, we should clarify how subtle may be differences between two molecular structures so that we can say that they are two conformers.'

It may be good to comment a little on the third possible conformer. Is there any NOE peak (or some other evidence) that can be attributed to a third conformer?

Author Response

Comments 1:It is understood in the field of NMR spectroscopy, that the presence of minor peaks in the spectra is often attributed to minor conformations which are often not taken into consideration. Thus, it is good to see the authors comment on 2 conformations. However, the authors also mention the possible presence of a third dominant conformer (lines 236-238):' We have chosen acetonitrile due to this fact and also because cyclosporin forms only two or three dominant
conformers in this solvent', although this seems to be in contradiction to Lines 265-266. 'Second, we should clarify how subtle may be differences between two molecular structures so that we can say that they are two conformers.'
Response 1: NOE’s in the Hα-Hα region include a peak labelled Mva11**–Bmt1** in Figure S6. However, it is impossible to follow the whole peptide backbone using HMBC and NOESY spectra due to the low population of this conformer. Also, there might be a conformer with all trans-bonds, which cannot be recognized so easily.
Regarding the contradictory statements in the text, we decided to rewrite it (line 144). The word “dominant” was removed (it’s a nonsense to say “three dominant conformers”). But in the phrase “how subtle may be differences between two molecular structures” we do not mean the two highest-populations conformers, so we have rewritten it also to make less ambiguous (line 176-177).